# Application of a regularised Coulomb sliding law to Jakobshavn Isbræ, West Greenland

Matt Trevers[1], Antony J. Payne[2], and Stephen L. Cornford[1]

[1]Centre for Polar Observation and Modelling, School of Geographical Sciences, University of Bristol, Bristol, UK
[2]School of Environmental Sciences, University of Liverpool, Liverpool, UK

**Correspondence:** Matt Trevers (matt.trevers@bristol.ac.uk)

**Abstract.** Reliable projections of future sea level rise from the polar ice sheets depend on the ability of ice sheet models to accurately reproduce flow dynamics in an evolving ice sheet system. Ice sheet models are sensitive to the choice of basal sliding law, which remains a significant source of uncertainty. In this study we apply a range of sliding laws to a hindcast model of Jakobshavn Isbræ, West Greenland from 2009 to 2018. We demonstrate that a linear viscous sliding law requires the assimilation of regular velocity observations into the model in order to reproduce the observed large seasonal and inter-annual variations in flow speed. This requirement introduces a major limitation for producing accurate future projections. A regularised Coulomb friction law, in which basal traction has an upper limit, is able to reproduce more accurately the range of speeds from 2012 to 2015, the period of peak flow and maximal retreat, without the requirement for assimilating regular observations. Additionally, we find evidence that the speed at which sliding transitions between power-law and Coulomb regimes may vary spatially and temporally. These results point towards the possible form of an ideal sliding parameterisation for accurately modelling fast-flowing glaciers and ice streams, although determining this is beyond the scope of this study.

## 1 Introduction

The form of the parameterisation of basal sliding is a key source of uncertainty in model projections of sea level rise from the polar ice sheets. Recent modelling studies have demonstrated strong sensitivity of the evolution of ice sheets to the choice of sliding law (e.g. Joughin et al., 2010a; Brondex et al., 2017; Nias et al., 2018), with more non-linear behaviour leading to higher sea level contributions (Ritz et al., 2015). Brondex et al. (2019) showed that projections of mass loss in the Amundsen Sea Embayment, West Antarctica, were highly sensitive to the choice of basal sliding law. Many ice sheet models have employed simple power-law relationships between sliding speed and basal traction (e.g. Krug et al., 2014; Cornford et al., 2015), following the sliding mechanism proposed by Weertman (1957). However, this mechanism fails to account for widespread cavitation resulting from high basal water pressure, which imposes an upper limit on the basal traction (Iken, 1981; Schoof, 2005; Gagliardini et al., 2007).

Joughin et al. (2019) demonstrated that the acceleration of the central trunk of Pine Island Glacier, West Antarctica, since 2002 in response to applied thinning and grounding line retreat was most accurately reproduced with a regularised Coulomb

friction law, in which the transition between power-law sliding (i.e. without cavitation) and Coulomb sliding (with cavitation) occurs above a threshold velocity. Gillet-Chaulet et al. (2016) had previously assimilated velocity observations for Pine Island Glacier between 1996 and 2010 to show that the observed acceleration was consistent with small values of the power-law index, indicating plastic deformation. De Rydt et al. (2021) calculated the optimal spatial distribution of the power-law index for Pine Island Glacier to explain the speedup and found that large regions of the central trunk upstream of the grounding line required a plastic bed. Similarly, Hillebrand et al. (2022) demonstrated that the recent speedup of Humboldt Glacier in North Greenland could be explained better with smaller values of the power-law index, producing more plastic behaviour. Coulomb behaviour had previously been demonstrated empirically for deformable subglacial tills (Tulaczyk et al., 2000) and the law for soft-bedded glaciers has been derived by Zoet and Iverson (2020). These laws take equivalent forms, raising the prospect for a potential universal sliding law (Minchew and Joughin, 2020).

Situated on the west coast of Greenland (Figure 1a), Jakobshavn Isbræ (JI) is one of the fastest flowing outlet glaciers of the Greenland Ice Sheet (GrIS), draining approximately 7% of the ice sheet area (Csatho et al., 2008). It has undergone dramatic changes over the last few decades, which are summarised in Section 1.1.

## 1.1 Recent evolution of Jakobshavn Isbræ

Until the late 1980s JI was in a stable configuration with limited seasonal ice front motion (Echelmeyer and Harrison, 1990) and a 15 km ice tongue, which was confined by the fjord walls and partially grounded near its terminus (Echelmeyer et al., 1991). The ice tongue thinned and disintegrated between 1996 and 2003 (Thomas, 2004), triggering significant ice front retreat, thinning and acceleration (e.g. Joughin et al., 2008; Howat and Eddy, 2011). Terminus flow speeds doubled following the ice tongue disintegration (Joughin et al., 2004) and a multi-kilometre annual cycle of ice front advance and retreat was established alongside significant seasonal velocity variations (Luckman and Murray, 2005).

Figure 2 shows how the velocity and ice front position of JI evolved from 2009 to 2018. Up until the winter of 2008 to 2009 a transient winter ice tongue formed from a mélange of fully and partially detached icebergs bonded by sea ice. From the winter of 2009 to 2010, this winter tongue stopped forming, which was concurrent with a reduction in the rigidity of the sikkusak in front of the terminus (Joughin et al., 2020). Supplementary Figure S1 shows the change from 2009 to 2010 from Landsat 7 images. The loss of the winter tongue explains the limited winter ice front advance after 2009. The ice front attained its furthest retreated positions in the summers from 2012 to 2015. The fastest flow speeds, in excess of 18 km yr$^{-1}$, were recorded in 2012 and 2013, after which there was some stagnation of flow speeds. After 2016 the terminus thickened and advanced, accompanied by ice flow deceleration back to speeds similar to pre-2012 (Khazendar et al., 2019). Additionally the winter ice tongue was observed to form again in this period (Joughin et al., 2020). The additional retreat and acceleration from 2012 to 2015 is thought to have been triggered by an incursion of warmer water into Illulisat Icefjord reaching the ice front, while the post-2016 re-advance was associated with a cooling of fjord waters driven by a multi-year cooling of the North Atlantic sub-polar gyre (Khazendar et al., 2019; Joughin et al., 2020).

The very fast flow speeds and high seasonal variability present a challenge for ice sheet models, making it an ideal test case for comparing the performance of sliding laws. Previous modelling studies applying linear or power law sliding parameterisations have failed to capture the peak flow speeds and magnitude of variability (e.g. Vieli and Nick, 2011; Bondzio et al., 2017). The mechanisms driving and controlling the retreat and acceleration of JI have already been the focus of many studies (e.g. Truffer and Echelmeyer, 2003; Thomas, 2004; van der Veen et al., 2011; Vieli and Nick, 2011; Joughin et al., 2012; Bondzio et al., 2017; Guo et al., 2019; Trevers et al., 2019).

In this study we apply a range of sliding laws to a hindcast model of JI from 2009 to 2018 forced by explicitly driving the ice front along its observed trajectory, and compare their ability to accurately reproduce the evolving flow state of the glacier during this period. In Section 3.1 we show that the regular assimilation of velocity observations into the model, achieved through a time-series inverse model (Section 2.1.3), is required by commonly used Weertman-like sliding laws in order to reproduce the observed variability in flow speeds. This is a major limitation for producing accurate projections of future behaviour. We demonstrate that a regularised Coulomb friction sliding law, in which a threshold fast-sliding speed governs the transition between power-law and Coulomb behaviour (Section 2.1.1), is more accurately able to reproduce the large variability in flow speeds observed for JI, and in Section 4.1 we discuss the mechanism driving the improved performance. In Section 3.2 we test the effect of varying the fast-sliding speed parameter in the regularised sliding law. Our results suggest the possibility of spatially and temporally varying parameters in the regularised law (Section 4.2).

## 2 Methods

### 2.1 BISICLES ice sheet model

In this study we use BISICLES (Cornford et al., 2013), a vertically-integrated ice flow model which is based on the L1L2 model of Schoof and Hindmarsh (2010). BISICLES features block-structured Adaptive Mesh Refinement (AMR) which enables fine grid resolution at the grounding line or ice front and coarse resolution elsewhere. The maximum grid spacing was 1200 m and 3 levels of refinement were applied to give a minimum spacing of 150 m.

Assuming that ice is in hydrostatic equilibrium, for bedrock elevation $b$ and ice thickness $h$, the upper surface $s$ is defined as

$$s = \max\left[h + b, \left(1 - \frac{\rho_i}{\rho_w}\right)h\right],\tag{1}$$

where $\rho_i$ and $\rho_w$ are the ice and ocean water densities respectively. The horizontal velocity $\boldsymbol{u}$ and ice thickness $h$ satisfy the mass conservation equation

$$\frac{\partial h}{\partial t} + \nabla \cdot [\boldsymbol{u}h] = \dot{M}_s - \dot{M}_b,\tag{2}$$

and the stress-balance equation

$$\nabla \cdot [\phi h \bar{\mu} (2\dot{\boldsymbol{\epsilon}} + 2\mathrm{tr}(\dot{\boldsymbol{\epsilon}})\,\mathbf{I})] + \boldsymbol{\tau}_b = \rho_i g h \nabla s, \tag{3}$$

together with appropriate boundary conditions. $\dot{M}_s$ and $\dot{M}_b$ are the surface accumulation and basal melt rates respectively, $\dot{\boldsymbol{\epsilon}}$ the horizontal strain-rate tensor, and $\mathbf{I}$ the identity matrix. The vertically-integrated effective viscosity $\phi h \bar{\mu}$ is calculated by integrating

$$\phi h \bar{\mu}(x,y) = \phi(x,y) \int_{s-h}^{h} \mu(x,y,z)\,\mathrm{d}z \tag{4}$$

between the ice base and surface. The ice sheet was subdivided into 10 vertical layers of equal thickness. The viscosity $\mu(x,y,z)$ satisfies

$$2\mu A(T)\left(4\mu^2 \dot{\boldsymbol{\epsilon}}^2 + |\rho_i g (s-z) \nabla s|^2\right) = 1, \tag{5}$$

in which $n = 3$ is the flow rate exponent from Glen's flow law and the Arrhenius factor $A(T)$ is dependent on the ice temper-
100 ature $T$ following Hooke (1981),

$$A(T) = A_0 \exp\left(\frac{3f}{[T_r - T]^k} - \frac{Q}{RT}\right), \tag{6}$$

where $A_0 = 0.093\ \mathrm{Pa}^{-3}\ \mathrm{yr}^{-1}$, $f = 0.16612\ \mathrm{K}^k$, $k = 1.17$, $T_r = 273.39\ \mathrm{K}$, $Q = 7.88 \times 10^4\ \mathrm{J\ mol}^{-1}$ is the creep activation energy and $R = 8.314\ \mathrm{J\ mol}^{-1}\ \mathrm{K}^{-1}$ is the universal gas constant. The ice stiffening factor $\phi(x,y)$ accounts for uncertainty in the ice temperature, ice fabric variations and macroscopic damage, and is estimated by solving an inverse problem (Section 2.1.2).
Low values of $\phi$ correspond to soft ice, which deforms more readily, while higher values correspond to more viscous ice, which exhibits greater horizontal stress transmission through membrane stresses.

### 2.1.1  Sliding laws

BISICLES implements a choice of sliding laws for calculating the basal traction $\boldsymbol{\tau}_b$. A power law with the form

$$\boldsymbol{\tau}_b = \begin{cases} -C\,|\boldsymbol{u}_b|^{m-1}\,\boldsymbol{u}_b & h\frac{\rho_i}{\rho_w} > -b \\ 0 & \text{otherwise}, \end{cases} \tag{7}$$

is commonly used in ice sheet models (e.g. Krug et al., 2014; Cornford et al., 2015). The role of effective pressure is subsumed into the value of the friction coefficient $C(x,y)$, which is determined empirically through an inverse method (see Section 2.1.2). The index $m = 1$ for linear viscous sliding, or $m = 1/3$ for a Weertman sliding law (Weertman, 1957), which is often used to model sliding over a hard bedrock. For finite $m$ there is no upper limit on the basal traction as the sliding speed increases. We also apply a regularised Coulomb friction sliding law of the form

$$\boldsymbol{\tau}_b = \begin{cases} -C\,|\boldsymbol{u}_b|^{m-1}\left(\frac{|\boldsymbol{u}_b|}{u_0} + 1\right)^{-m}\boldsymbol{u}_b & h\frac{\rho_i}{\rho_w} > -b \\ 0 & \text{otherwise}. \end{cases} \tag{8}$$

This is equivalent to the regularised Coulomb law introduced by Joughin et al. (2019) but expressed such that the units of $C$ match those of the power law for equivalent $m$. The fast-sliding speed $u_0$ subsumes the role of basal effective pressure, about which we have limited knowledge. $u_0$ is assumed to be constant through the domain. $|u| \gg u_o$ produces perfectly plastic behaviour where the basal traction is independent of the sliding speed, while $|u| < u_o$ tends towards power law behaviour. Sliding law profiles for a range of parameter values are shown in Supplementary Figure S2.

### 2.1.2 Inverse method

In BISICLES, optimised fields of $C(x,y)$ and $\phi(x,y)$, which limit the misfit between modelled and observed ice flow speeds, are computed by an inverse method. We choose values of $C$ and $\phi$ that minimise a cost function

$$J = J_m + J_p, \tag{9}$$

comprising a misfit function

$$J_m = \frac{1}{2} \int_\Omega \alpha_u^2(x,y) \left(|\boldsymbol{u}_{\mathrm{mod}}| - |\boldsymbol{u}_{\mathrm{obs}}|\right)^2 \mathrm{d}\Omega \tag{10}$$

and a penalty function

$$J_p = \frac{\alpha_C^2}{2} \int_\Omega |\nabla C|^2 \, \mathrm{d}\Omega + \frac{\alpha_\phi^2}{2} \int_\Omega |\nabla \phi|^2 \, \mathrm{d}\Omega, \tag{11}$$

across the model domain $\Omega$ using a nonlinear conjugate gradient method (Cornford et al., 2015, Appendix B1). $\boldsymbol{u}_{\mathrm{mod}}$ and $\boldsymbol{u}_{\mathrm{obs}}$ are the modelled and observed velocities respectively and $\alpha_u^2(x,y)$ takes the value 1 where velocity data exist and 0 elsewhere. The Tikhonov regularisation coefficients $\alpha_C^2$ and $\alpha_\phi^2$ are necessary for two purposes. Firstly, the cost function $J$ has no unique minimum with respect to $C$ and $\phi$: the problem is under-determined, since we seek two unknown fields using only one field of data. Secondly, the inverse problem is sensitive to small variations in $\boldsymbol{u}_{\mathrm{obs}}$, in other words it is ill-conditioned. Smoothing resulting from the regularisation filters out the effect of noise in $\boldsymbol{u}_{\mathrm{obs}}$ in the final $C$ and $\phi$ fields. The choice of $\alpha_C^2$ and $\alpha_\phi^2$ represents a compromise between low values, which produce a very close match to the observations but potentially result in overfitting to noise in the input data, and high values, which produce smooth fields but a larger misfit. Optimal values of $\alpha_C^2 = 1 \times 10^{1.5}$ and $\alpha_\phi^2 = 1 \times 10^{7.5}$ were found using an heuristic L-curve method following Hansen and O'Leary (1993). We set

$$C_0 = \begin{cases} \min\left(\frac{\rho_i g h |\nabla s|}{|\boldsymbol{u}_{\mathrm{obs}}| + 1 \times 10^{-6}}, 1 \times 10^5\right) & \alpha_u^2 > 0 \text{ and } h\frac{\rho_i}{\rho_w} > -b \\ 20 & \text{otherwise} \end{cases} \tag{12}$$

and $\phi_0 = 1$ as initial guesses for $C$ and $\phi$ respectively. The inverse problem is insensitive to the choice of sliding law since it is effectively optimising $\tau_b$, so linear viscous sliding (Equation 7, $m = 1$) was applied.

### 2.1.3 Time-series inverse model

When observations from multiple epochs are available, the inverse method in Section 2.1.2 can be extended to regularise in time between successive observations through additional terms in the cost function (Equation 9),

$$J = J_m + J_p + \chi_C^2 J_{C,t} + \chi_\phi^2 J_{\phi,t} \tag{13}$$

where

$$J_{C,t} = \int_\Omega \ln\left(\frac{C(x,y,t)}{C(x,y,t-\Delta t)}\right) d\Omega \tag{14}$$

$$J_{\phi,t} = \int_\Omega \ln\left(\frac{\phi(x,y,t)}{\phi(x,y,t-\Delta t)}\right) d\Omega. \tag{15}$$

$\chi_C^2$ and $\chi_\phi^2$ are temporal regularisation coefficients. The time-series inverse model essentially consists of a time-series of individual inverse models with unique geometry and velocity inputs, with the temporal regularisation constraining variation in $C$ and $\phi$ between successive snapshots. In practice we use a single observation with good spatial coverage as a reference timeslice, and the resulting $C_{\text{ref}}$ and $\phi_{\text{ref}}$ form the initial guesses $C_0(t)$ and $\phi_0(t)$ for each subsequent timeslice. The temporal regularisation therefore enables the time-series inverse model to infer values of $C$ and $\phi$ in locations with gaps in the observational data that are too large for the spatial regularisation to cover. The purpose of the time-series inverse model is to produce the temporally evolving inputs of $C$ and $\phi$ for the LV_TRANS hindcast model (see Section 2.2.3).

## 2.2 Experimental setup

### 2.2.1 Model data

Model inputs are shown in Figure 3 for the ice front and ice stream, or in Supplementary Figure S3 for the full domain. The model domain covers an area 518.4 km by 384 km in extent, encompassing the full JI drainage basin. Bedrock topography at 150 m resolution was provided by BedMachine v3 (Morlighem et al., 2017). 40 unique surface DEMs were constructed for each quarter year in the study period by iteratively summing annual rates of surface elevation change to the Greenland Ice Mapping Project (GIMP) surface DEM (Howat et al., 2014), which has a nominal date of 2007. Annual surface elevation change rates were provided by Khan et al. (2016) for 2007 through to 2011, and by Khan et al. (2022) for 2011 onwards. Auto-delineated ice fronts from Zhang et al. (2019) were applied to the surface DEMs, with the furthest advanced ice front for each quarter being selected.

A map of mean velocity for 2008 and 2009 with good coverage across the entire JI drainage basin (Rignot and Mouginot, 2012, v4), was used for the initial reference timeslice. A time-series of 40 quarter-yearly mean velocity maps was compiled from a range of products from the MEaSUREs project (Joughin et al., 2010b, 2018a). Datasets NSIDC-0478 (v2, Joughin et al., 2018b), NSIDC-0727 (v3, Joughin, 2021b) and NSIDC-0731(v3, Joughin, 2021c), derived from TerraSAR-X, Sentinel-1 and LandSat 8 observations, provided velocities across the drainage basin for each quarter as available. Additionally, high

resolution 11 day TerraSAR-X velocity maps, using a combination of speckle-tracking and interferometry (NSIDC-0481, v3, Joughin, 2021a), provided observations for the fast-flowing ice stream and ice front, with all available observations within the quarter year mean period averaged. $\alpha_u^2$ was set to 2 where TerraSAR-X observations (NSIDC-0481, v3, Joughin, 2021a) were

available, 1 where observations were available from other MEaSUREs products and 0 where no observations exist. A combination of spatial (Section 2.1.2) and temporal (Section 2.1.3) regularisation was able to infer values of $C$ and $\phi$ for regions without observations. Quarterly mean flow speeds and individual TerraSAR-X measurements at selected locations are shown in Figure 2. Supplementary Table S1 summarises the data sources used to construct the time-series of quarterly inputs for the time-series inverse model. All datasets were resampled onto the 150 m BedMachine grid.

A three-dimensional temperature field with 10 uniformly spaced vertical layer was provided by a 50,000 year thermodynamical spinup using the modern ice sheet geometry and velocity for 2008 to 2009, carried out previously using BISICLES (Trevers, 2021). Geothermal heat flux from Shapiro and Ritzwoller (2004) and surface air temperature from Ettema et al. (2009), with an additional component of 5 °C at the coastal boundary of the domain linearly reducing to 0 °C at the ice divide boundary,

were taken as boundary conditions for the temperature spinup. The thermodynamical spinup accounted for both horizontal and vertical thermal advection as well as vertical heat diffusion. The temperature was assumed not to evolve during the study period from 2009 to 2018.

### 2.2.2   Model initialisation

The model inversion procedure was carried out for the mean 2008 and 2009 reference velocity (Rignot and Mouginot, 2012,

v4) and ice geometry produced for the first quarter of 2009 (2009-Q1). The resulting reference fields $C_{\text{ref}}$ and $\phi_{\text{ref}}$ are shown in Figure 3e and 3f.

A time-series model inversion was then carried out for each of the 40 unique quarterly timeslices of velocity and geometry described in Section 2.2.1. The additional temporal regularisation terms of Equation 13 were applied, with $C_{\text{ref}}$ and $\phi_{\text{ref}}$

used as the initial guess for $C$ and $\phi$ for each timeslice. A time-series of mean values is shown in Supplementary Figure S4 and maps of the difference of $C$ and $\phi$ for each quarterly timeslice relative to 2009-Q1 are shown in Supplementary Figures S5 and S6. Low values of $C$ and $\phi$ in the middle of the study period correspond to the fastest sliding between 2012 and 2015, while increasing values correspond to flow stagnation from after 2016.

We relaxed the 2009-Q1 geometry for 50 years in order to produce an ice sheet surface consistent with the flow field and to reduce ice flux divergence anomalies (Seroussi et al., 2011). The thickness of floating ice and the positions of the grounding line and ice front were held fixed, while grounded ice was allowed to freely evolve. The mean surface accumulation rate from 1960 to 1989 from RACMO2.3p2 (Noël et al., 2018) was applied. Finally, the inverse model procedure was repeated for the relaxed geometry to match the 2009-Q1 velocities to produce the initial state for hindcast model runs. Note that the relaxation

and additional inversion were only performed for the 2009-Q1 timeslice.

The model inversions and relaxation were carried out using a linear viscous sliding law (Equation 7, $m = 1$), therefore the resulting fields of $C$ are only applicable for linear viscous models. Unique fields of $C$ for alternative sliding laws were calculated by equating optimised values of $\tau_b$ in the relevant expressions. This ensures that initial ice thickness and velocities are equal between simulations. Supplementary Figure S7 shows that differences in ice flux divergence between simulations are small relative to the magnitude of ice flux divergence early in the experiments.

### 2.2.3 Hindcast model

The hindcast experiments were run as prognostic models from the start of 2009 to the end of 2018. Experiments were initiated with the 2009-Q1 relaxed surface geometry and $C$ and $\phi$ inputs (Section 2.2.2). Annual surface mass balance rates for each year from RACMO2.3p2 (Noël et al., 2018), supplied at 1 km resolution and resampled onto the 150 m BedMachine grid, were applied while the ice temperature was assumed to remain constant. The ice front was driven along the smoothed observed trajectory (Figure 2b) by calculating the calving rate required to generate the required amount of advance or retreat at each timestep. Along the flowline (Figure 1b), the calving rate $u_C{}^*$ was calculated for the modelled ice flow speed measured at the intersection of the flowline and ice front ($u_T{}^*$). Elsewhere, the calving rate $u_C$ was scaled according to the modelled ice front velocity $u_T$,

$$u_C\left(x,y\right) = u_T\left(x,y\right)\frac{u_C{}^*}{u_T{}^*}. \tag{16}$$

Thus, whilst the terminus is only directly driven along the observed trajectory where it intersects with the central flowline, the entire ice stream front advances and retreats in step with it.

A novel scheme was used to determine the rate of surface elevation change. At the ice front, the surface elevation was allowed to evolve freely. At locations further than 15 km from the ice front, the rate of surface elevation change was prescribed according to the observed annual elevation change rates described in Section 2.2.1. Up to 15 km from the ice front, a linearly graduated mixture of free surface and prescribed elevation change rates were applied. This scheme was applied to limit differences in surface elevation between models applying different sliding laws. Supplementary Figure S7 shows that differences in ice flux divergence at the start of the hindcast experiments are small relative to the magnitude of ice flux divergence.

### 2.2.4 Experiments

Four hindcast model experiments were run to compare different sliding laws. Two experiments applied the linear viscous sliding law (Equation 7 with $m = 1$). In one of these experiments (LV_STAT) the 2009-Q1 $C$ and $\phi$ inputs were applied, and in the other experiment (LV_TRANS) the full quarterly time-series of $C$ and $\phi$ inputs determined from the time-series inverse model were applied sequentially with linear temporal interpolation between inputs to ensure a smooth transition. Two other experiments applied a Weertman sliding law (WE_STAT, Equation 7, $m = 1/3$) and a regularised sliding law (Equation 8, $m = 1/3$) with $u_0 = 500$ m yr$^{-1}$ (RC_500_STAT). Both WE_STAT and RC_500_STAT experiments applied static 2009-Q1

$C$ and $\phi$ inputs used throughout the experiment in each case. Table 1 contains details of these experiments.

| Experiment | Sliding law | $m$ | $u_0$ (m yr$^{-1}$) | $C$ and $\phi$ inputs |
|---|---|---|---|---|
| LV_STAT | Power law (Equation 7) | 1 (Linear viscous) | n/a | 2009-Q1 non-evolving |
| LV_TRANS | Power law (Equation 7) | 1 (Linear viscous) | n/a | Quarter-yearly time-series |
| WE_STAT | Power law (Equation 7) | 1/3 (Weertman) | n/a | 2009-Q1 non-evolving |
| RC_500_STAT | Regularised law (Equation 8) | 1/3 | 500 | 2009-Q1 non-evolving |

**Table 1.** Details of sliding law comparison experiments.

A further set of hindcast model experiments were carried out in which the regularised sliding law was applied with a range of values of $u_0$ from 500 m yr$^{-1}$ to 10000 m yr$^{-1}$. Static 2009-Q1 $C$ and $\phi$ inputs were again applied throughout each experiment.

## 3 Results

### 3.1 Sliding law comparison

Figure 4 shows the modelled ice flow speeds at site M0 for the sliding law comparison experiments (see Supplementary Figure
S8 for more detail). The mean percentage errors at M0 are 13.9%, 6.0%, 13.6% and 16.7% for the LV_STAT, LV_TRANS, WE_STAT and RC_500_STAT experiments respectively. When considering only the central period from 2012 to 2015 when peak sliding speeds and ice front retreat were observed (grey shading in Figure 4) the mean percentage errors are 11.4%, 5.7%, 13.2% and 6.8%. The LV_TRANS experiment therefore performed best over both the full duration of the experiment and from 2012 to 2015. RC_500_STAT also performed very well between 2012 and 2015, accurately reproducing both the peak sum-
mer speeds as well as the winter deceleration. Supplementary Figure S8c shows that RC_500_STAT reproduced the seasonal variability best between 2012 and 2015, but outside of this period it overestimated both the magnitude and variability of flow speeds. The LV_STAT and WE_STAT experiments both failed to reproduce the peak flow speeds between 2012 and 2015, and also underestimated the seasonal variability throughout the experiment. Before 2012, the WE_STAT and RC_500_STAT experiments both overestimated flow speeds. RC_500_STAT significantly overestimated the seasonal variability during 2009, but
produced accurate variability through 2010 and 2011 while overestimating the flow speeds. All experiments except LV_TRANS failed to account for the deceleration from 2016 onwards. We attribute this to more significant winter sikkusak formation in Illulisat Icefjord after 2016 (Joughin et al., 2020), which is not accounted for in the model physics. By contrast, LV_TRANS was able to reproduce the observed slower sliding speeds since it assimilated these observations.

At sites situated further upstream from the ice front, errors in the RC_500_STAT and LV_TRANS experiments increased while the error decreased in the LV_STAT and WE_STAT experiments (Figure 5b). This results from a bias towards overesti-mating sliding speeds which worsens with distance upstream (c.f. Supplementary Figure S9 at M15 vs S8 at M0), causing the

RC_500_STAT and LV_TRANS experiments to overestimate flow speeds throughout the experiment. At all sites upstream of M0 the LV_TRANS experiment reproduced the seasonal variability most accurately, with the error in RC_500_STAT increasing more rapidly than in the other laws (Figure 5c).

## 3.2 Fast-sliding speed

Mean percentage errors and annual range percentage errors from 2012 to 2015 across the range of $u_0$ are presented in Figure 6. Time-series of flow speeds at all sites are also shown in Supplementary Figure S10. $u_0 = 2000$ m yr$^{-1}$ performed slightly better than 500 m yr$^{-1}$ at M0 but the relative slopes suggest that 500 m yr$^{-1}$ would win out downstream of M0. The same tendency towards increasing overestimation of flow speeds with distance upstream was seen in these experiments (Supplementary Figure S10). The percentage error increased with distance upstream for smaller values of $u_0$ but decreased for 5000 m yr$^{-1}$ and 10000 m yr$^{-1}$. The behaviour of $u_0 = 5000$ m yr$^{-1}$ and 10000 m yr$^{-1}$ was more similar to the Weertman law than for smaller values of $u_0$. $u_0 = 10000$ m yr$^{-1}$ produced the smallest mean percentage errors at sites further upstream than 3 km, but overall it performed worst at reproducing the seasonal variability.

## 4  Discussion

### 4.1  Sliding law comparison

The fields $C$ and $\phi$ account for various unknowns in the state of the ice sheet, which aren't explicitly described in the model physics. $C$ subsumes the effects of unknown substrate type, uncertainties in topography at multiple scales, basal ice temperature and water pressure at the bed. $\phi$ accounts for uncertainty in ice temperature, fabric and macroscale damage. A complete knowledge of the properties of the ice and bedrock and a full description of the physics affecting ice flow would render these model inputs unnecessary. Numerical modelling studies generally assume that the properties parameterised by these fields do not change significantly during the course of a model run (e.g. Cornford et al., 2015; Bondzio et al., 2017). However, Habermann et al. (2013) performed repeated model inversions for JI at intervals between 1985 and 2008 and found a lowering of effective basal yield stress over that period, and Joughin et al. (2012) also found a reduction in basal traction between the 1990s and 2009. De Rydt et al. (2021) showed that acceleration of Pine Island Glacier in West Antarctica from 1996 to 2016 could not be modelled by glacier geometry changes alone and required changes to the rheological or basal properties.

The linear viscous sliding law does not account for the effects of cavitation or changing basal effective pressure. The significant underestimation of peak summer flow speeds in LV_STAT demonstrate that thinning and changes in buttressing resulting from ice front motion alone are insufficient to fully resolve the flow dynamics. The enhanced ability of LV_TRANS to reproduce the changes in flow speed is to be expected since the velocity observations assimilated at regular intervals are effectively being reproduced by the model. Changes in basal or englacial properties are accounted for by the evolving values of $C$ and $\phi$. Choi et al. (2023) similarly found that using transient friction and viscosity coefficients in a numerical model of Kjer Glacier,

Greenland, increased the accuracy of modelled velocities compared with using static coefficients. This produces an accurate hindcast but the reliance on assimilating regular observations is problematic for using the model to perform projections of the future evolution of JI.

The regularised sliding law (RC_500_STAT) was able to accurately reproduce the peak flow speeds and variability at M0 between 2012 and 2015 without requiring transient $C$ and $\phi$ inputs. The improvement of the regularised law can be understood by considering the basal traction distributed across a region rather than just at a single location. We define the Grounding Zone (GZ), shown in Figure 1b, as a box around site M0 covering several square kilometers, which is convenient for analysis since grounding lines retreat and advance across it during simulations. Figure 7 compares $\tau_b$ averaged across the GZ against the total grounded ice area in the GZ for the LV_STAT and RC_500_STAT experiments. Changes in grounded area occur as the grounding line moves backwards and forwards within the GZ, with varying rates of thinning between the experiments accounting for differences in grounded area. For any individual cell, ungrounding of the ice results in zero basal traction and a tendency to accelerate, inducing acceleration in nearby cells. For the linear viscous sliding law (LV_STAT), grounded cells experience an increase in basal traction proportional to their acceleration, limiting the overall regional reduction in traction and hence constraining the acceleration. By contrast, the regularised law (RC_500_STAT) basal traction is only weakly dependent on sliding speed for fast sliding (Supplementary Figure S2). Therefore the increase in traction for grounded cells is limited, and the regional traction is strongly dependent on the ratio of grounded to floating ice. As a result the acceleration resulting from grounding line retreat is less constrained. In other words, the stress transfer in the linear viscous law is local, whereas the regularised law is able to balance the loss of basal traction in one location with a more non-local transmission of stress. This mechanism is clearly demonstrated by the strong linear correlation between traction and grounded area (Figure 7, bottom row) as compared with the linear viscous law (top row), which has a significant velocity dependency. In this way, the regularised law is better able to adjust flow speeds in response to the loss of traction as ice thins and comes afloat. Our explanation of this mechanism lends support to Minchew et al. (2019) in their rebuttal of Stearns and van der Veen (2018). Whilst it may be the case that $\tau_b$ is insensitive to $u_b$ at a given location, which is consistent with the regularised law, our results demonstrate that the magnitude of basal traction distributed across a wider area is an important control on the sliding speed.

## 4.2 Fast-sliding speed

Supplementary Figure S11 shows that the $R^2$ value relating grounded area and basal traction within the grounding GZ decreases with increasing fast-sliding speed $u_0$. This relationship explains why smaller values of $u_0$ performed best between 2012 and 2015 when the grounding line was in the vicinity of M0. The results from further upstream, where larger values of $u_0$ produced a better match to observations, might point to a spatially varying value of $u_0$, with the optimal value possibly determined by some function of distance from the grounding line. This can be explained by considering that $u_0$ controls the transition in the form of the sliding law, from power law behaviour at speeds slower than $u_0$, to Coulomb behaviour above $u_0$ where the sliding speed is effectively decoupled from the basal traction. Note that the transition between regimes occurs smoothly and does not occur abruptly at $u_0$ (Figure 8). Coulomb or even near-plastic ($u_0 = 0$ m yr$^{-1}$) behaviour is the dom-

inant regime close to the grounding line, therefore a low value of $u_0$ is optimal here. Further upstream away from the direct influence of the grounding line, flow follows power-law behaviour instead. Whilst power-law behaviour can be modelled with the regularised law, a higher value of $u_0$ is required here since flow at these sites is still significantly faster than 500 m yr$^{-1}$. Figure 8 shows that $u_0 = 500$ m yr$^{-1}$ produces near-plastic sliding behaviour for the 2012 to 2015 range of sliding speeds at both M0 and M15. Conversely $u_0 = 10000$ m yr$^{-1}$ still produces some sensitivity of $\tau_b$ to $u_b$ at both sites, and is close to Weertman-like behaviour at M15. This illustrates how a regularised law with spatially varying $u_0$ may enable the simulation of a range of basal rheologies for different regions of the ice sheet.

Our results are consistent with De Rydt et al. (2021) who showed that the acceleration of Pine Island Glacier between 1996 and 2016 could best be explained with a spatially varying value of the sliding exponent $m$ (equivalent to our $1/m$), including wide regions with large $m$ indicating effectively plastic bed conditions beneath the fast-flowing central valley. They did not use a regularised law, instead applying a typical power law. The regularised law is able to accommodate both power-law and Coulomb-plastic behaviour without requiring a varying value of $m$, because the transition between these regimes is governed by $u_0$. The results of De Rydt et al. (2021) could therefore likely be replicated with a homogeneous value of $m$ and spatially varying $u_0$. In practical terms there may be little difference between the two approaches, but we suggest that a regularised law with varying $u_0$ may be a more natural way to model the dynamics since $u_0$ governs the transition between different behaviors.

Spatial heterogeneity in the value of $u_0$ could be attributed to a variety of potential physical processes related either to characteristics of the underlying bed or to the geometry of the glacier. For a soft bed with a saturated till, $u_0$ may represent the point at which the till starts to deform, which is dependent on the size and spacing of clasts embedded in the till (Zoet and Iverson, 2020). For hard-bedded sliding, lower values of $u_0$ may indicate variations in bed morphology that facilitate cavitation at slower speeds (Joughin et al., 2019; Helanow et al., 2021; Woodard et al., 2022). While there is evidence that JI is underlain by a deformable till (Block and Bell, 2011; Habermann et al., 2013; Shapero et al., 2016), the regularised Coulomb law is applicable in the case of both hard and soft-bedded sliding (Minchew and Joughin, 2020; Helanow et al., 2021). $u_0$ also partially (along with the basal friction coefficient $C$) subsumes the role of effective pressure (Schoof, 2005; Joughin et al., 2019), which varies spatially depending on the ice thickness, ocean connectivity (Parizek et al., 2013) and subglacial hydrology. Our knowledge of the water pressure at the bed is limited, hence the optimisation of $C$ accounts for the effect of unknown spatial variations in effective pressure. However we would expect the effective pressure to change as the ice thins towards flotation, which a non-evolving $C$ cannot account for. In Section 4.1 we argued that the regularised law is better able to account for changes in traction resulting from grounding line motion. This suggests that spatial variations in $u_0$ may be optimal for modelling the dynamics of JI. When considering the longer-term evolution, more significant grounding line motion may also entail temporal variations in $u_0$. We note that considering temporal variations in $u_0$ may invoke similar limitations to those that applied to the LV_TRANS model. Considering only ice thickness and ocean connectivity should lead us to expect lower effective pressure and hence $u_0$ closer to the terminus. Subglacial hydrology may have more complex spatial distribution as well as a temporal component related to seasonally varying meltwater inputs and evolution of the hydrological system over the course of a year

(e.g. Tedstone et al., 2013). These different factors indicate that $u_0$ may be difficult to parameterise and may instead need to be determined empirically.

## 5 Conclusions

A linear viscous sliding law with non-evolving model inputs cannot accurately model the evolution of JI from 2009 to 2018 since it is unable to accommodate the very high seasonal velocity variations that are observed in the fast flowing regions of the ice stream. Assimilating regular velocity observations to produce transient inputs of basal friction coefficient and ice stiffening factor significantly improved the model's ability to calculate the full range of velocities. However this strategy renders the model reliant upon regular velocity observations, which are unavailable for future modelling applications. We explored the use of different sliding laws with non-evolving inputs to address this limitation. A regularised Coulomb friction sliding law, which accounts for the effect of widespread cavitation due to high basal water pressure, reproduced velocities most accurately between 2012 and 2015 when velocities reached their peak and variability was greatest. Although we applied a uniform value of the fast-sliding speed ($u_0$), which controls the transition between power-law and Coulomb sliding regimes, our results suggest that the value of this threshold may vary both spatially and temporally. This suggests that improved projections of the future evolution of fast-flowing ice streams may be achieved by employing a regularised sliding law with spatially varying parameters.

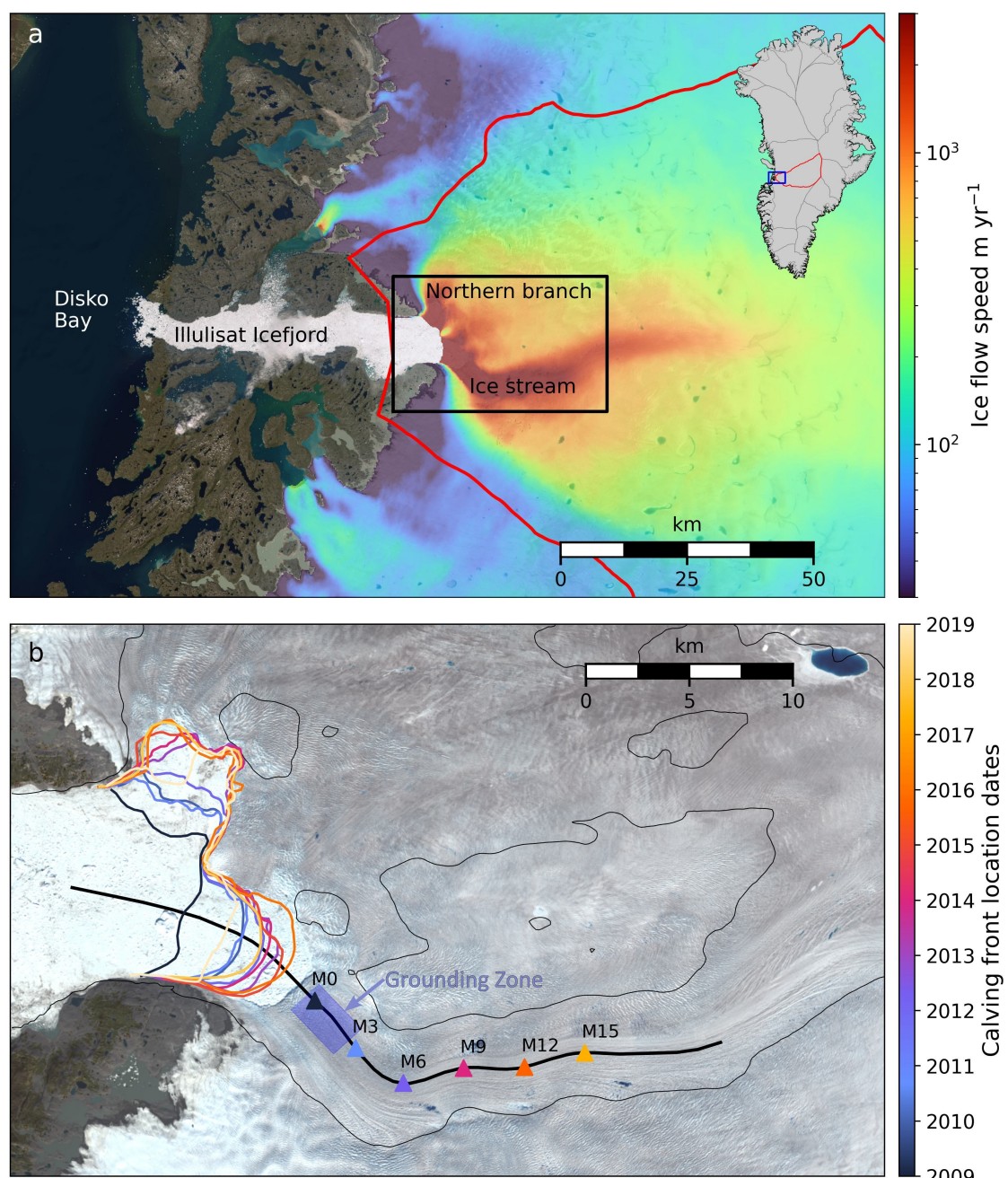

**Figure 1.** The JI study region situated in West Greenland. (a) shows the ablation zone and ice stream, with 2008 – 2009 ice velocities from Rignot and Mouginot (2012, v4). Inset map in (a) shows drainage basins from Ekholm (1996) with the JI basin highlighted in red, and the blue box defining the extent of panel (a). (b) shows detail of the black box in (a). Year start ice fronts from 2009 to 2019 (colored solid lines) were manually delineated from SAR intensity images of Lemos et al. (2018). Sites M0 to M15 are highlighted. The shaded blue region is the Grounding Zone (GZ) which is used for further analysis in Secion 4.1. Thin black lines delineate the sea level bedrock elevation contour. Background images in (a) and (b) were captured by Landsat 8 on August 9th, 2016.

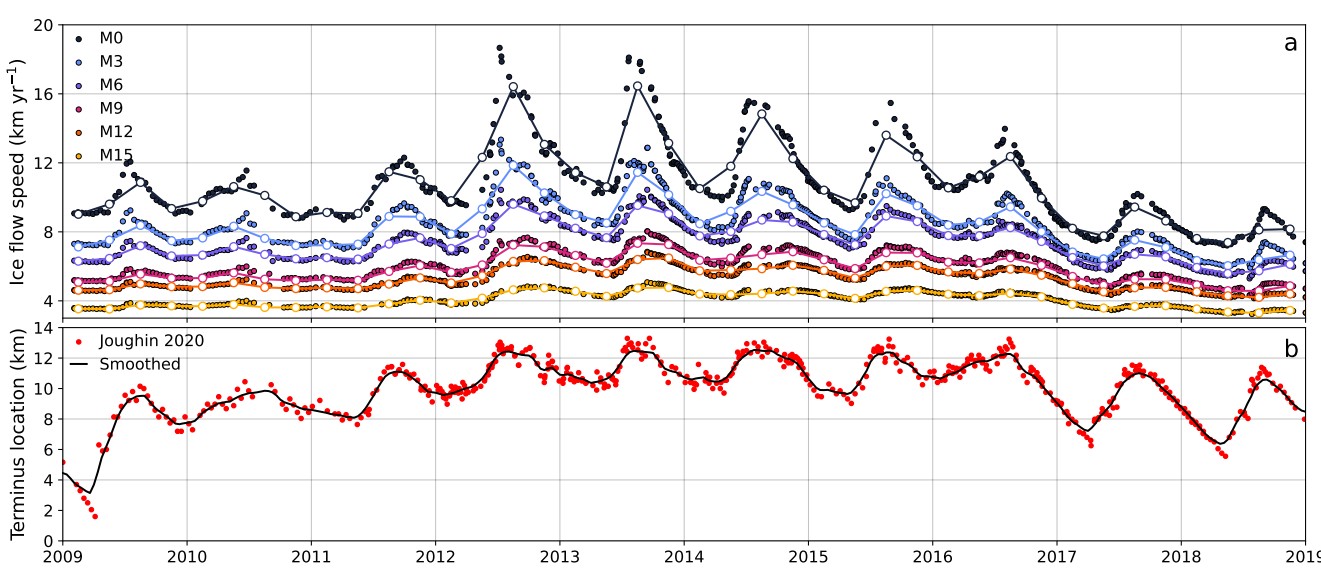

**Figure 2.** (a) Ice flow speed time-series measured at sites M0 to M15. Small scatter points are individual measurements extracted from TerraSAR-X/TanDEM-X velocity time series provided by Joughin et al. (2020). Connected open circles denote quarterly mean flow speeds at these locations. (b) Ice front locations measured along a flowline, with a smoothed time-series in black.

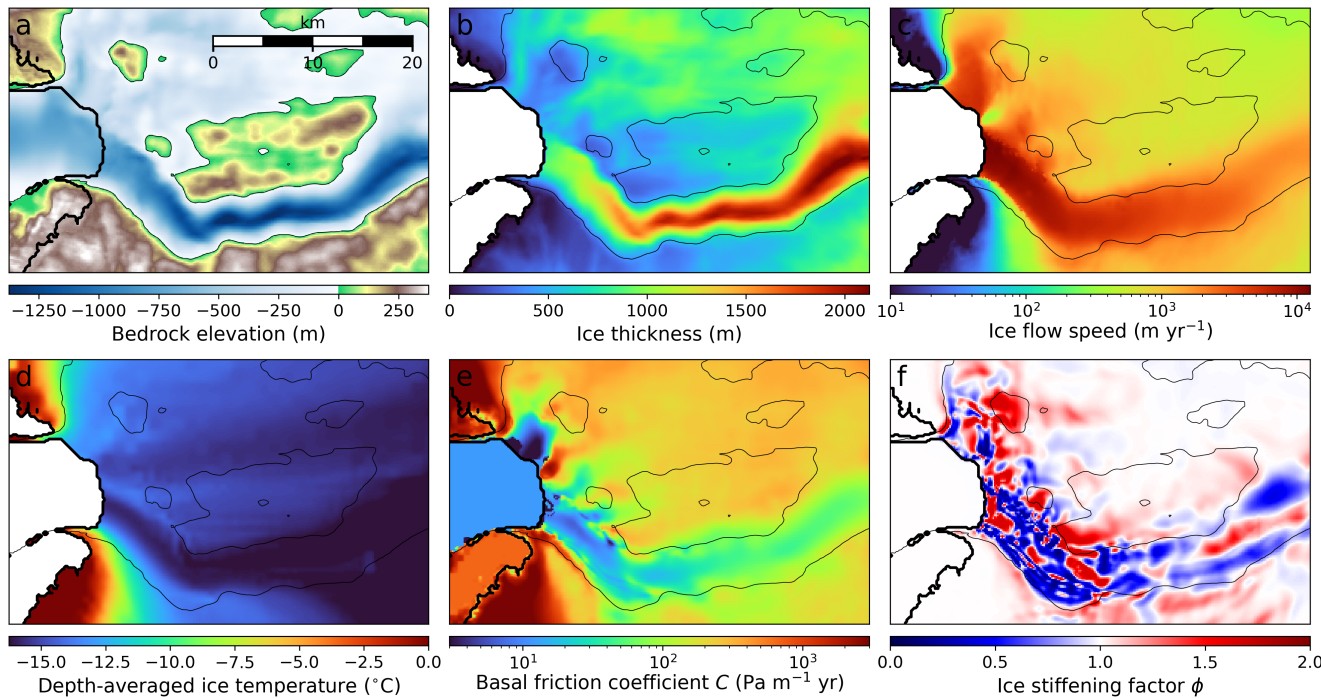

**Figure 3.** Model inputs. (a) Bedrock topography (BedMachine v3, Morlighem et al., 2017), (b) ice thickness (BedMachine v3, Morlighem et al., 2017), (c) 2008 and 2009 mean flow speed (Rignot and Mouginot, 2012, v4), (d) depth-averaged ice temperature, (e) reference basal friction coefficient $C_{\mathrm{ref}}$ and (f) reference ice stiffening factor $\phi_{\mathrm{ref}}$. Thin contours delineate the sea level bedrock elevation contour while the thick black line marks the ice extent. The region shown is the same as in Figure 1b, and not the full model domain (see Supplementary Figure S3).

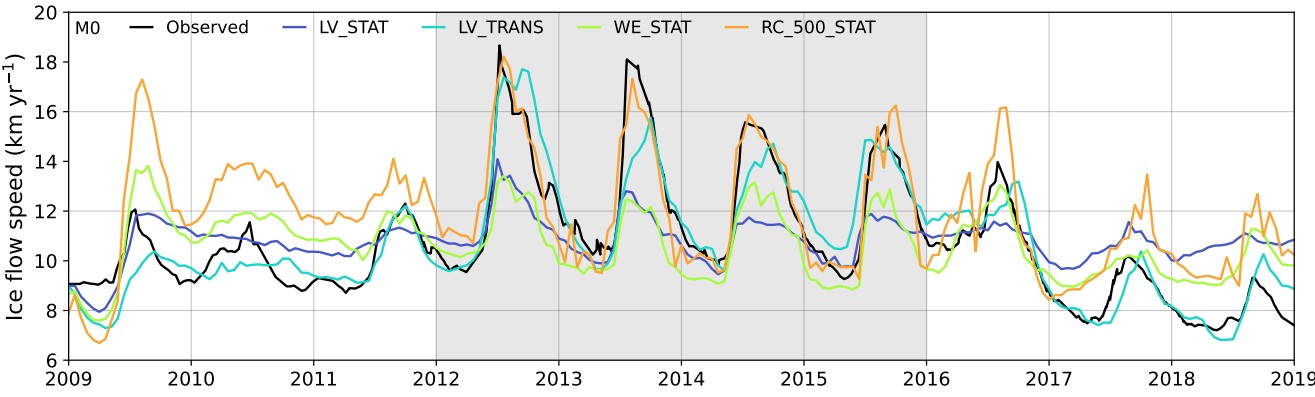

**Figure 4.** Observed and modelled flow speeds at site M0 in the sliding law comparison hindcast model experiments. See Supplementary Figure S8 for additional detail. The grey-shaded region covers the four-year period when observed flow speeds peaked and calving fronts attained maximal retreat. This period is referenced in Figures 5 and 6.

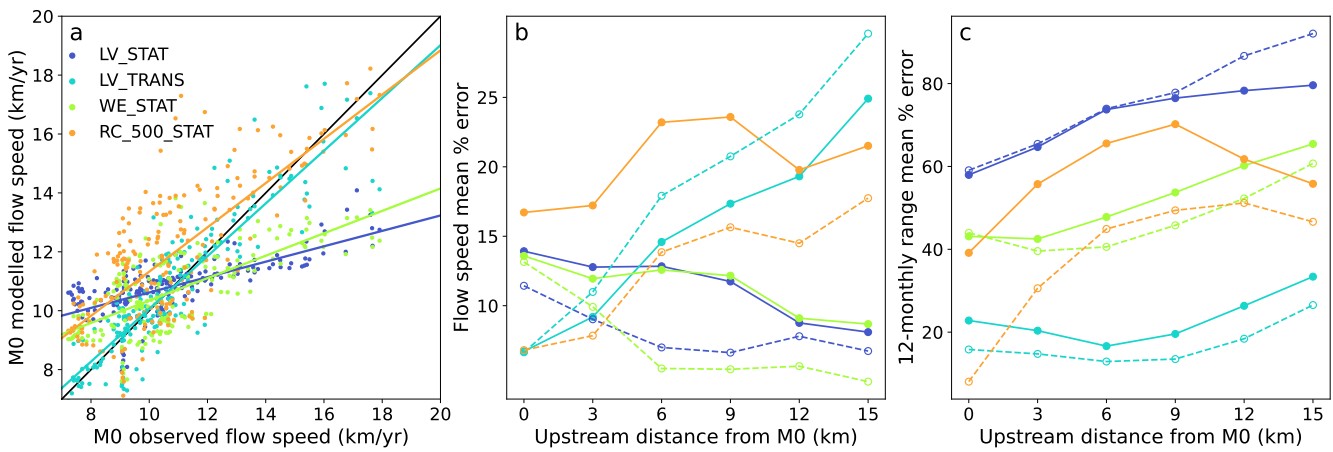

**Figure 5.** (a) Scatter plots of modelled versus observed flow speeds at M0 for the experiments in Figure 4, with lines of best fit included. The diagonal black line indicates a perfect match to observations. (b) Mean percentage error in modelled flow speed measured at each site. (c) Mean percentage error in modelled 12-monthly range measured at each site. The 12-monthly range is calculated as the difference between maximum and minimum flow speeds within 6 months of the measurement. In (b) and (c) filled circles connected by solid lines are calculated for the full period while open circles connected by dashed lines are calculated for the period from 2012 to 2015 (grey shading in Figure 4).

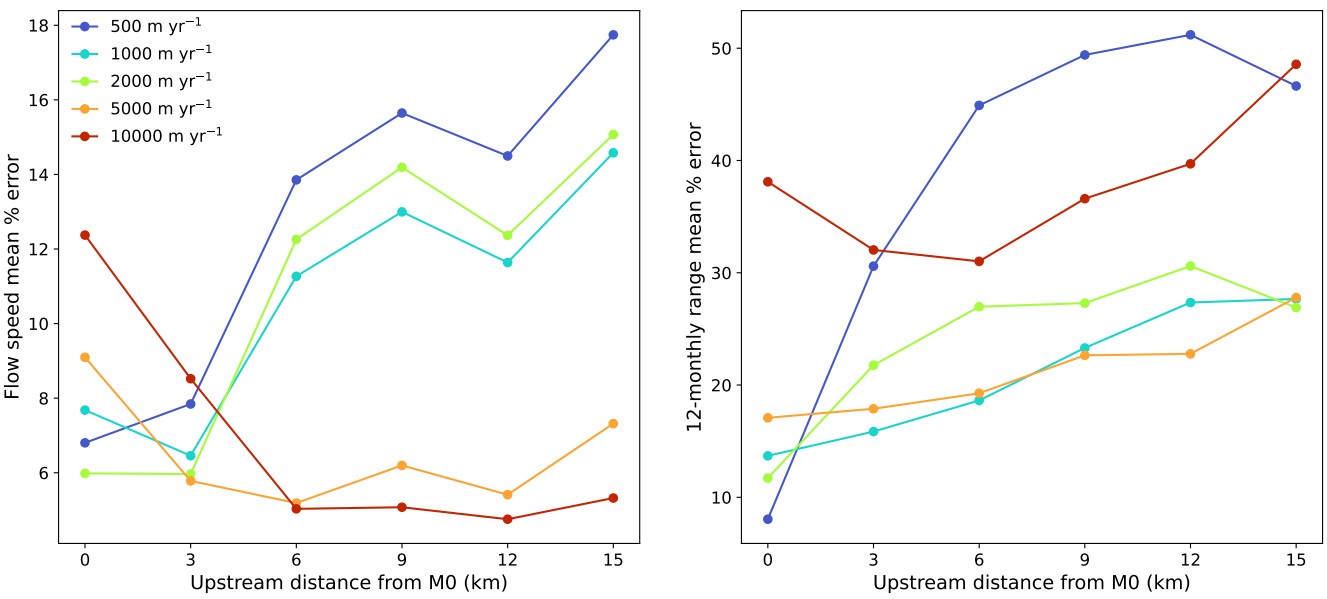

**Figure 6.** (a) and (b) as in Figure 5 (b) and (c) respectively, for the regularised sliding law with a range of values of $u_0$. All values are calculated for the period from 2012 to 2015 (grey shading in Figure 4).

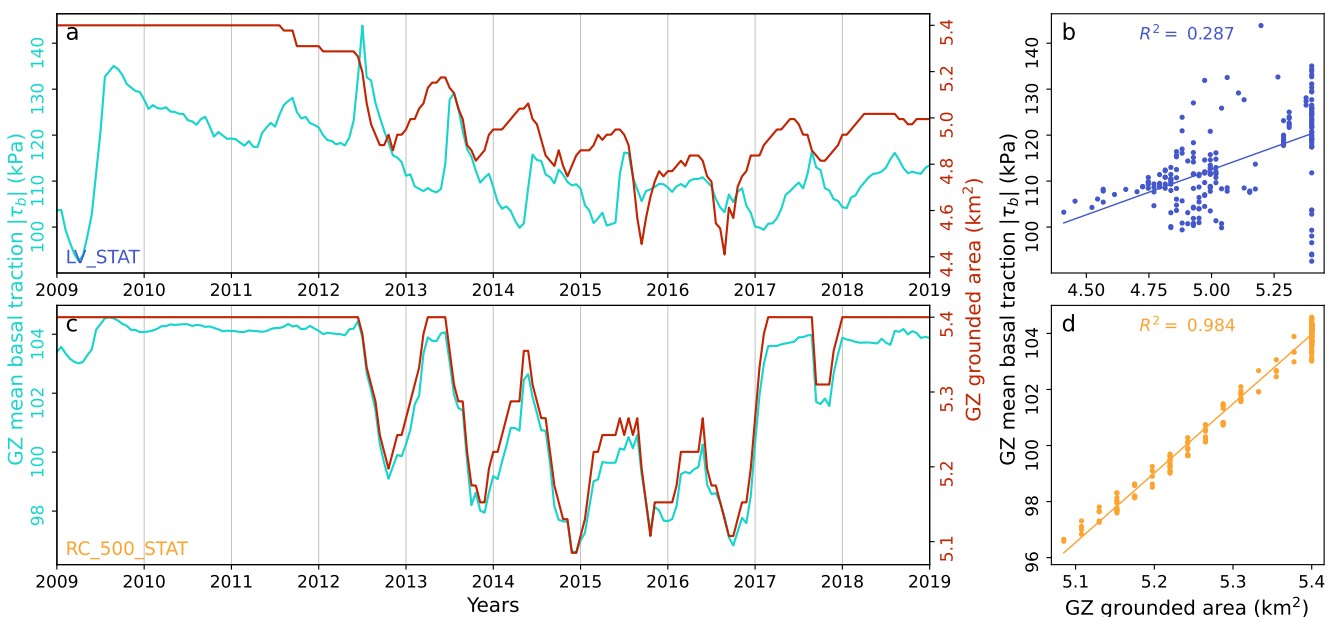

**Figure 7.** Comparison of basal traction and grounded area for different sliding laws. Top row: LV_STAT experiment. Bottom row: RC_500_STAT experiment. (a) and (c) time-series of the mean basal traction $\tau_b$ across the GZ (blue) and the GZ grounded area (red). (b) and (d) present the same data as (a) and (c) but as scatter plots with corresponding lines of best fit and R-squared values.

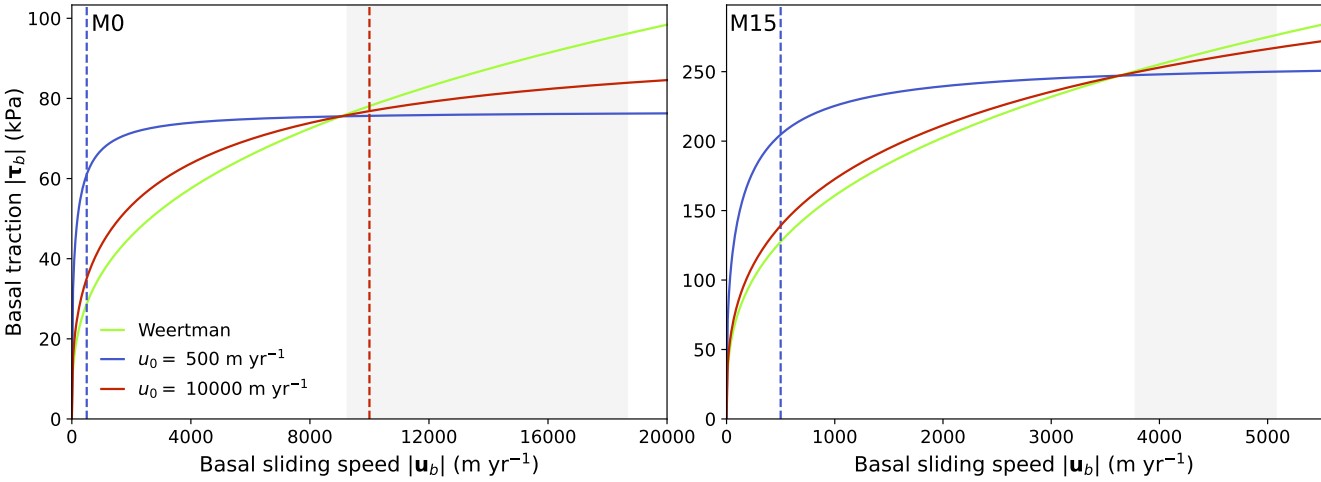

**Figure 8.** Basal traction calculated at M0 (left panel) and M15 (right panel) for the Weertman law (green) and regularised law with $u_0$ values of 500 m yr$^{-1}$ (blue) and 10000 m yr$^{-1}$ (red). Vertical dashed lines indicate $u_0$ values for the corresponding regularised law curves, although the scale for M15 (right panel) excludes $u_0 = 10000$ m yr$^{-1}$. Shaded grey regions indicate the 2012-2015 range of observed flow speeds at each site.

*Code availability.* The open-source BISICLES ice flow model is available for download from https://github.com/ggslc/bisicles-uob. All datasets used in this study have been cited. Model outputs and scripts are available from the authors upon request.

*Author contributions.* MT devised this study, carried out all experiments and analysis and prepared the manuscript with contributions from AJP and SLC.

*Competing interests.* The authors declare that they have no conflict of interest.

*Acknowledgements.* MT was supported by a NERC GW4+ Doctoral Training Partnership studentship from the Natural Environment Research Council (NE/L002434/1) and also by the NSFGEO-NERC Pliocene sea level amplitudes (PLIOAMP) project (NE/T007397/1). We thank Jacob Woodard, Stephen Price and the Editor for their insightful and constructive comments, which have substantially improved the quality of this manuscript.

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
