# Peer review of "Application of a regularised Coulomb sliding law to Jakobshavn Isbræ, West Greenland"

_EGUsphere, 2024_

## Author Comment (AC1)

**Application of a regularised Coulomb sliding law to Jakobshavn Isbræ, West Greenland**

The Cryosphere Discussions, https://doi.org/10.5194/egusphere-2024-1040

Matt Trevers, Antony J. Payne, and Stephen L. Cornford
Correspondence: Matt Trevers (matt.trevers@bristol.ac.uk)

**Response to Referee Comment #1 (**https://doi.org/10.5194/egusphere-2024-1040-RC1**)**

Referee comments are in black, author responses are in blue, *suggested changes are in italic blue*.
* * *
Trevers et al. test several glacier sliding laws to see how well they reproduce the flow dynamics of Jakobshaven Isbrae gathered from 2009 to 2018. Specifically, they compare the more widely used Weertman and linear sliding laws to a regularized Coulomb sliding law. Theory suggests that the regularized Coulomb sliding law can account for basal cavitation and heterogeneous bed materials important for controlling glacier slip whereas the former two sliding laws cannot. Both the Weertman and linear sliding laws produce very poor model results. To improve model performance, Trevers implements an active reparameterization scheme that allows the model to update its parameters with changes in the glacier's velocity field. While this reparameterization scheme improves model performance, its reliance on the velocity data prevent its use for any ice flow projections. In contrast, the regularized Coulomb sliding law can generally reproduce the variable sliding velocities at Jakobshaben Isbrae without reparameterization. The manuscript's subject matter is of general interest to the earth science community as improving the sliding law parameterization is pivotal for accurately forecasting future sea-level rise. I found the manuscript to be well written and their conclusions to be well supported by their results. I had a few suggestions that I think would clarify some points, all of which are relatively minor. However, I am enthusiastic about this manuscript being published and think it is a significant contribution to the field.

We would like to thank the reviewer for taking the time to read and review our manuscript, and for their insightful, constructive and encouraging remarks.

General comments:

I think some clarification is needed for understanding what experiments you carried out and why. For instance, the suffix 'TRANS' was confusing for the model experiments that did not vary the C and phi parameters. I would emphasize why you are allowing the LV model to vary these parameters. My understanding is that it is to show that the RC model can produce essentially the same model fit without having to use detailed velocity data. This allows the RC model to be used more effectively for projections. I think this point is really important, but it wasn't obvious during my first reading. I would suggest setting up the problem a bit better in the introduction and throughout the manuscript so that this impressive result isn't saved until the discussion.

Thank you for spotting this mistake. 'STAT' and 'TRANS' are intended to refer to non-evolving and evolving C/phi parameters respectively. This crept in as a typo when we changed the experiment naming convention, and is certainly confusing for the reader.

*We will correct 'WE_TRANS' and 'RC_500_TRANS' to 'WE_STAT' and 'RC_500_STAT' wherever they occurred throughout the document.*

It is unclear to me how you are defining the 'grounding zone'. More of the glacier is grounded beyond the box. I think my confusion here can be resolved with a change in word choice of grounding zone or clarification on what you mean by grounding zone. It follows that I was unsure how you determined the grounded area. Please elaborate on this in the text.

The Grounding Zone (GZ) is a box that we have drawn around site M0 encompassing several square km. It is convenient for analysis since the grounding line retreats and advances across it during hindcast simulations. The Grounded Area is then simply calculated from the number of cells inside the GZ with grounded ice.

*We will keep the name 'Grounding Zone' but we have added some additional text to the paragraph where it is first mentioned in Section 4.1 to more clearly define it. We will also add a label to Figure 1b, and reference Section 4.1 in the Figure 1 caption.*

Specific comments:

Abstract – It is unclear from the abstract if you assimilate velocity data with the other sliding laws. I also think you can better setup the problem that you are trying to solve with these experiments to really drive home the importance of the work you've done here.

We agree that the abstract did not do a good job of priming the reader for the aims and message of the study.

*We will add text to the abstract to explicitly link the regular assimilation (time-series inverse model) with the linear viscous sliding law, and an additional sentence to highlight that this is a limitation which the regularised law is able to overcome.*

Section 1 - I think it's worth pointing out here that the form of the hard bedded and soft bedded slip laws is similar. Otherwise, I don't think your point about a universal slip law makes sense.

We agree that the equivalence of the hard and soft bedded laws is important to highlight.

*We will split up the final sentence of paragraph 2 of Section 1, and add a clause to the beginning of the sentence to highlight the equivalence.*

Section1 – continued - I think here is where you should also setup the problem and talk the reader through your hypothesis and how you are setting about to prove it. That is, lay the foundation for the importance of the experiments you are going to run in the paper.

We agree that we could do more to set up the problem addressed by the study.

*Along with the aforementioned changes to the abstract, we will add a paragraph to the end of Section 1.1 to outline the problem, the limitation in the linear viscous law and how we show the improvement with the regularised law.*

Line 52 – Maybe too detailed, but I was interested in if we know why change in water circulation happened.

There are complicated oceanographic drivers of this. It's beyond the scope of this study to explain them, but Khazendar et al. (2019) contains much more detail.

*We will briefly mentioned that these are linked to cooling of the North Atlantic sub-polar gyre.*

Line 127 – Why?

*The inverse method is effectively optimising the basal stress so it should be insensitive to the sliding law used. It is therefore easier to just perform it once for a single sliding law rather than having to rescale $\alpha_c$ in Equation 11 to achieve the same thing.*

*We will extend the final sentence of this paragraph to include this reasoning.*

Line 142 – What is the resolution of BedMachine?

*150 m resolution. We will include this detail.*

Line 143 – Please define GIMP.

*Greenland Ice Mapping Project. This detail will be included*

Line 158 – How did you make the different datasets compatible spatially? Did you do any resampling?

*Yes, all datasets were resampled onto the 150 m BedMachine grid. This will be mentioned at the end of this paragraph*

Line 166 – I would suggest moving this to the beginning of the next paragraph.

*We will move this sentence to the beginning of the first paragraph of Section 2.2.1, where we feel it is most appropriate.*

Line 192 – What is the temporal and spatial resolution of RACMO?

*The data were provided at annual temporal resolution and 1 km spatial resolution, resampled on the BedMachine grid. We will include this detail.*

Line 210 – I found the name scheme here a bit confusing. I thought the 'STAT' or 'TRANS' part of the names related to static to transient evolutions of C and phi. Is that incorrect? Also, I would put the name of the Weertman model here since you did it for all the others.

*As mentioned above, the use of 'TRANS' instead of 'STAT' here was in error, and this will be corrected everywhere in the manuscript. We will also clarify the sliding law names in Table 1, and the experiment name for the Weertman law will be mentioned in line 208.*

Line 210 – Why didn't you use a transient C and phi with the Weertman and RC models? Again, I think this goes to better setup the problem you're trying to solve.

*The point was to demonstrate that the linear viscous model required the transient inputs in order to accurately reproduce the observed velocities, but that this is a problematic modelling strategy. The RC model was able to overcome these limitations without requiring transient inputs. We hope that the changes already mentioned above will address this question and prep the reader better on the motivation for this study.*

Line 226 – Please explain somewhere why you didn't establish an initial state with the different flow laws.

*Upon evaluation, we realised that the explanation we had provided for the poor performance of RC_500_STAT early in the experiment was not satisfactory. We will remove this explanation. We aren't able to provide an alternative satisfactory explanation, although we note that RC_500_STAT only performs really badly in 2009, after which it reproduces the annual range*

quite well during 2010 and 2011, while still overestimating the after flow speeds. *We will highlight this.*

We are content that it is legitimate to use the same relaxed state for all experiments, since the initial velocities are equal in all experiments, and differences in the flux divergence between simulations early on are small relative to the magnitude of the flux divergence (see Supplementary Figure S7). *We will add a sentence to justify this to the end of Section 2.2.2.*

Line 229 – Please elaborate on this last point. At first, I didn't quite follow why the inverse model would allow the LV_TRANS model to perform better after 2016.

This point wasn't phrased very clearly. LV_TRANS isn't capturing any physics that the other models aren't, simply it's assimilating the slower velocities and then reproducing them.

*We will change the text of this sentence to clarify this point.*

259-267 – This is well explained here but I felt like the impact of this statement could have been setup better in the introduction and results. It took me reading it a few times to realize why the RC model was so much better even though it doesn't look much different from the LV_TRANS model results.

We hope that the aforementioned changes to the abstract and introduction will help to prep the reader better on the significance of this explanation.

274 – I'm having a difficult time understand how you calculated the grounded area. Are you somehow accounting for cavities or is it just the percent of the glacier that is not floating above the "grounding zone".

The grounded area is simply calculated from the number of cells within the GZ that contain grounded ice. However, this sentence omitted the detail that grounded area is calculated within the GZ.

*We will include this detail here. We will also update the Figure 7 caption accordingly*

305 –Somewhere in this section I would suggest explaining why it is better to be able to vary u_o over *m*.

In practice there may be little difference, but we suggest that a spatially varying u_0 may be a more natural way to model this because u_0 governs the transition between regimes.

*We will add a sentence to the end of this paragraph to make this argument.*

310 – Fast-sliding speed is u_o correct? Please just use the symbol once you define it earlier on. You can define it again in the section heading or early in this section. I was a bit confused at first because you first introduce u_o with a lot of other variables and I quickly forgot the meaning of this specific one. I think putting parenthesis or commas around the symbols would also help the readers follow the definitions of these variables.

We agree that this is confusing for the reader. *We will use u_0 instead throughout this paragraph.*

314 – Woodard et al., 2023 also talks about this.

Woodard JB, Zoet LK, Iverson NR, Helanow C. Inferring forms of glacier slip laws from estimates of ice-bed separation during glacier slip. *Journal of Glaciology*. 2023;69(274):324-332. doi:10.1017/jog.2022.63

*This will be included as a reference.*

Section 5 – I found this whole paragraph to be difficult to follow. Please consider rewriting. I'll put a few specific issues I had below.

324 – LV can but it needs to be re-parameterized with velocity data. Maybe merge the first and second sentences to avoid confusion.

*We won't merge the sentences but we will include a qualifier to say that this is for non-evolving inputs.*

331 – Unclear to me what you mean by transition speed here. Is this u_o?

*Yes. We will change this.*

Figure 5 – I had a hard time seeing the colors in the legend. Consider making the points larger.

*We will increase the point size, and the colour scheme will also be changed.*

Figure 7 – I had a hard time seeing the yellow in these plots. Especially the axis text. Please consider changing the color.

*We will change the colour scheme for this figure, along with Figures 4 to 8 and S8 to S10 to keep them in line with the same colour scheme. This colour scheme should be clearer and still be colour-blindness friendly.*

Figure S1 – I could not tell from this where the tongue was. Consider outlining the tongue to help orient the readers. A north arrow I think would also help here.

*We agree that it's not easy to tell where the glacier is grounded or afloat.*

*We will change the images in this figure so that they're closer to a year of separation. We will also outline the glacier terminus for both years, add a North arrow, and change the figure caption as well.*

---

## Author Comment (AC2)

**Application of a regularised Coulomb sliding law to Jakobshavn Isbræ, West Greenland**

The Cryosphere Discussions, https://doi.org/10.5194/egusphere-2024-1040

Matt Trevers, Antony J. Payne, and Stephen L. Cornford
Correspondence: Matt Trevers (matt.trevers@bristol.ac.uk)

**Response to Referee Comment #2 (**https://doi.org/10.5194/egusphere-2024-1040-RC2**)**

Referee comments are in black, author responses are in blue, *suggested changes are in italics blue*.
* * *
In this manuscript, Trevers et al. explore how well a range of commonly used sliding laws – linear-viscous, Weertman, and Coulomb-friction – perform in a model at mimicking observed speeds along the main trunk of Greenland's Jakobshavn Glacier during the time period between 2009 and 2019. They find that, while no single sliding law with static (fixed in time) parameters does a good job of matching observed velocities for the entire time period, a regularized Coulomb-friction law does a much better job (relative to the others tested) at matching observations during the time period from 2012-2016, during which the glacier exhibited both the highest overall peak speeds and the largest range in annual speeds. The authors go on to discuss the reasons behind these different model behaviors, after which they hypothesize that a regularized, Coulomb-friction sliding law with a spatially variable regularization parameter may provide the optimal choice for best-matching observed speeds (without requiring time-dependent optimization, something that is obviously not a viable choice if the ultimate goal is to use the optimized model for projections of future change).

Overall, this is an informative paper that is well written and presents interesting and useful findings. It confirms and expands upon related findings from other recent work and will be appreciated by readers of The Cryosphere. Most of my comments below are minor and editorial in nature, aiming to improve the readability of the paper. I do think that the overall direction and findings of the paper could be more clearly hinted at and summarized up front. Because of the way the abstract and introduction are currently written, I was anticipating that the focus was going to be more on what was gained and learned from conducting a time-dependent initialization / assimilation of observations. As currently written, the conclusions may leave the reader a bit unsatisfied; having queued up the interest, we are left wanting to see the results of the proposed optimal sliding law, That is, a Coulomb-friction law with a spatially variable, optimized, regularization parameter. While I realize that may be beyond the scope of the current work, it could be nice to end with the proposal to conduct future work to this end, if that is indeed the author's ultimate intention.

We are glad that the reviewer found the manuscript to be interesting, and would like to thank them for the helpful and insightful review.

We agree that the current structuring of the abstract and introduction don't make it clear for the reader quite what the motivation and aim of the study are, with too much of a focus on the time-series inversion which isn't the main focus of the study. It is not our intention to carry out future work to investigate how to spatially optimise the fast-sliding speed.

*We will make some changes to the abstract to point that the time-dependent assimilation is required to make the linear viscous law work, and that this presents a limitation. We will*

*highlight that the regularised Coulomb law is able to reproduce the observed velocities without this requirement.*

*We will also add a paragraph to the end of Section 1.1 to lay out the motivation, goals and structure of the paper.*

Abstract

3: "...range OF sliding laws..."

5: "...the OBSERVED large seasonal and ..."

5: "... AND THAT the assimilation of regular ..." ("while" does not seem appropriate here since the suggestions regarding the sliding-law-type and the assimilation of velocities are two distinct and very different topics).

7: "was" -> "is" ? Note that the tense in most of the rest of the abstract is present, not past (e.g., in next sentence you say "we find" rather than "we found").

7: It might be more informative here to say " ... able to reproduce the range of speeds observed during the period of peak flow, from 20XX-20YY" (or something like this). After readying the full paper it seems to me that this is the more important and interesting conclusion to highlight.

7: "Finally, we find ..." (missing comma)

*We will make all these recommended changes*

Last sentence – maybe make this a bit more clear, e.g. if you are putting forth a proposal for such a sliding law as part of this contribution.

*We will add a clause to this final sentence to point out that this is beyond the scope for this study.*

Main Text

12-15: You might also add a reference here to Hillebrand et al., who conducted a somewhat similar set of exercises to try and model the behavior of Humboldt Glacier, Greenland, and found similar w.r.t. importance and sensitivity of power-law exponent in sliding law. There may also be some other findings from that paper w.r.t. sliding law types, param. values, calibration against observations, etc. that are relevant to / warrant some discussion here (The Cryosphere, 16, 4679–4700, 2022 https://doi.org/10.5194/tc-16-4679-2022).

21-20: As above, the findings from Hillebrand et al. may be relevant to discuss here.

*We will add the Hillebrand et al. reference at this point in the text.*

37: "... ice tongue, which ..." (comma after "which")

47-48: "...flow speeds, in excess of 18km/yr, ..." (missing commas?)

66: Maybe clarify "... block-structured mesh refinement ..."?

*We will change this.*

91: "...soft ice, which ..." and "... viscous ice, which ..." (comma after "which")

98: Again, comma after "which"? Note that I'll stop mentioning this explicitly from here on and just suggest the authors check the remaining manuscript and ensure this is used correctly and consistently throughout (some places in do use a comma after which and some do not).

*We will add commas at all the suggested locations, and will check the text for any other places where commas would be appropriate and consistent*

118: Move the colon forward? E.g., suggest "... with respect to C and phi: since we seek to unknown fields ..."

*We will rearrange this sentence*

Section 2.1.3: Did you try without the timeseries regularization? If so, what were the results? Or, put another way, would it make sense here to provide a bit of additional information on what motived this choice (as opposed to just taking the straight-up, best-fit optimization fields for every individual timeslice? I.e., it's not entirely clear what you get from the time-lagged portion of the optimization (unless I've misunderstood it).

The temporal regularisation enables the time-series inverse model to fill in locations where gaps in the data are too large for the spatial regularisation to cover, so long as the reference timeslice has good spatial coverage.

*We will add a sentence to clarify this. We will also add another sentence to highlight that the purpose of the time-series regularisation is just to produce the time-series of C/phi inputs for the hindcast model.*

142-145: Was there any independent check done on the accuracy of the DEMs constructed in this way? It seems like simply accumulating dh/dt year on year for many years in a row could also result in the accumulation of error. Alternatively, were any estimates made of the potential errors in the constructed DEMs and/or the potential impact on those errors on the simulated velocities?

Due to time constraints we did not carry out any checks or error/sensitivity analysis relating to this.

152: What does "as available" mean here? Were there 3 month periods for any given year where obs. vel. data were not available (and, if so, what was done to generate velocities for those time periods)?

This refers to the fact that the velocity datasets have gaps in.

*We will add a sentence to this paragraph to point out that the regularisation helps to fill in the gaps.*

152-154: Were the vels. for the faster moving trunk generated from feature tracking rather than interferometry? Perhaps in this section you could clarify if interferometry or feature tracking was used for the calculation of velocities (or if a combination of methods was used for velocities covering each scene at each time period).

These used a combination of speckle-tracking and interferometry. *We will include this detail.*

160-164: Clarify if the temperature spin-up is done as part of the BISICLES model or using some other model. And what sort of temperature model is used? Does it count for both horizontal and vertical advection? A few additional details would be appreciated.

*These details will be added to this paragraph.*

2.2.2: I'm confused about how the time-series optimization works. Is the same reference state always used (i.e., the first quarter of 2009) or is a new reference state – linked to the optimization from the prior quarter – used each time?

The same reference state is used for each quarterly timeslice. We did try running it sequentially, i.e. using the outputs from the previous timeslice as the initial guess for the next timeslice. We found that this tended to exaggerate the magnitude of variations in the resulting C and phi outputs, but did not substantially affect the resulting velocity fields, so we preferred to use the same reference state each time.

*This will be clarified in the text.*

179-180: "… AN ice sheet surface … and TO reduce …"

*This will be corrected.*

188: "…were calculated by equating Tau_b WITH ITS OPTIMZED VALUE (?) in the relevant expressions."

*This will be clarified in the text.*

179-184: It's unclear to me how the generated DEMs factor in here. If you relaxed the initial sfc elevation via forward modeling, then presumably that sfc is much different from that of the initial DEM. Did you accumulate anomalies from the DEM differences to your modeled sfc elevation? Were the DEMs just used for the optimization of the sliding coefficients, etc. …but then not necessarily consistent with the model ice sheet surface for those same time periods?

It's applied both in the timeseries inverse model to produce the surface elevations there, but then also as a prescribed thickness change rate in hindcast experiments upstream of the calving front (see below).

190-197: Is the calving front position is specified by observations or calculated? Initially here, it sounds like you calculate the calving rate required to match the observed ice front. But then below that you say that it's only the centerline that gets this treatment and the rest of the ice front (?) is scaled according to this rate and the local velocity. In that case, does any other ice front position than the centerline match the observed ice front position over time? A little additional clarification would help here.

Yes, this treatment is only applied along the central flowline. It would be significantly more complicated to apply this treatment everywhere along the calving front, or to multiple flowlines. Scaling the calving rate like this still enables the ice stream front to advance and retreat in step.

*We will add an extra sentence following Equation 16 to mention this.*

198-200: Perhaps this addresses my question above. It sounds like the observed thinning rates (inland of 15 km) are applied to the model sfc state. Is this in addition to or instead of any thinning that the model calculates for these locations? Is the mix of observed vs. modeled thinning rates between 15km and the terminus just a linear combination based on the distance along the flowline between the two regions?

The prescribed thinning beyond 15 km is instead of any modelled thinning. Essentially, an additional accumulation/ablation component is calculated to account for the difference

between the modelled thickness change and the prescribed thickness change. For positions for than 15 km from the calving front, this component is added in full. Less than 15 km, this component is partially added with a factor that scales linearly with distance from 0 at the calving front to 1 at 15 km.

207-208: Clarify – linear interpolation was done between the quarterly inputs determined from inversions?

*Yes. This will be clarified in the text.*

257: "and required the inference of changes in" ... could this just be "and required changes in"? If the inference part is important, then it seems like you may want to make the additional clarification in this sentence that you are talking about the modeling of these velocity changes (as opposed to the actual changes themselves).

*We will change "explained" to "modelled" to clarify this.*

261-263, 265-267: I think it could worth explaining more clearly and early on in the paper, when you first discuss the methods and different sliding laws used, that you are not advocating here for time-dependent optimization of basal slide parameters (since in some sense, as you point out here, this is "cheating" a bit). Rather, this is your baseline for clearly identifying if / that changes in basal sliding / basal traction are required to fit the observations. After that, your goal is to find the best sliding law and set of fixed parameters for that law that also allow you to best match the time-varying observations. This is something that you might also consider trying to clarify / prepare the reader for a bit better in the introduction and abstract.

We agree that the goal of the paper needs to be made clearer upfront. We hope that the changes already discussed to the abstract and introduction, as well as to the time-series inversion methods section, will provide more direction for the reader on the first time of reading.

270-282: Another way to think about this (or possibly explain it) is that ungrounding and loss of basal traction at one point has to be made up for at some other location (in order to obey force balance); the loss of support at the bed requires the transfer of stress, laterally or longitudinally, to another part of the bed. For a more linear sliding law, this transfer can be quite local (neighboring grid cells) but for a coulomb-friction law and a semi-uniform failure strength in a region (here, the grounding zone), this is not possible. That stress transfer will simply increase traction at neighboring regions, leading to failure of the bed there as well. In this sense, the Coulomb-friction law leads to / requires a non-local transfer of stress.

Yes, this is correct. *We will add a sentence to this section to point out the local/non-local stress transmission between the two laws.*

282-283: "Through this mechanism ... is able to account for in effective pressure without explicitly ...". Can you elaborate further on what you mean here? I'm not sure I follow exactly (and maybe other readers will have a similar problem).

On evaluation this was a poorly thought out sentence and a bit misleading.

*We will point out instead that the regularised law is accounting for the loss of traction when cells come afloat, but not to changes in effective pressure resulting from other changes in basal properties.*

290: "The results from further …". Something missing from / wrong with the beginning of this sentence that makes the rest of it confusing.

*This sentence was missing a clause, this will be added in to make the sentence make sense.*

292-293: Can you be explicit here if / that the transition between "slow" speeds and the faster speeds allowed by a Coulomb law is in fact given by the value of u_0 (i.e., if u_0 is 1000 m/yr, do we expect linear-viscous behavior below that speed and Coulomb behavior above it?).

*Yes. We will alter this sentence to make this a bit clearer. It should also be noted that the transition is smooth and doesn't occur abruptly at u_0. A sentence to point this out will be added.*

308-309: Practically speaking though, is there much difference? In each case, you would need to figure out / specify the spatially varying parameter field values. Do you think that a spatially varying u_0 is more physically reasonable / intuitive than a spatially varying m?

We agree that there may not be much difference in practice, but we suggest that the regularised law provides a more natural way to model the dynamics since it combines both regimes and the fast-sliding speed governs the transition. *We will add a sentence to make this argument.*

310-320: Be explicit here that by "the fast sliding speed" you mean the value of u_0 (?).

*This will be changed to use "u_0" throughout this section for clarity.*

Again, in 317 you say that (presumably) this value subsumes the role of effective pressure. Can you be clear about what you mean by that and why? Is this simply a mathematical / functional argument because it appears in the denominator as effective pressure often does (and so both u_0 and eff. pressure have an inverse relationship with basal traction)? Note that this may be the same question I'm asking about above for lines 282-283.

*We will add a few sentences here to flesh out this argument. Essentially, C and u_0 both partially subsume the effect of unknown spatial variations in the effective pressure. Effective pressure will change near the grounding line in response to thickening or thinning. A non-regularised law cannot account for these changes with a static value of C, but we argued in Section 4.1 that the regularised law can account for changes in traction resulting from grounding line motion. Therefore it might be optimal to have a spatially varying u_0, instead of subsuming all the effective pressure variation into C. Over longer periods with more significant grounding line retreat, temporal variation may also be necessary.*

328: "We explore the use of different sliding laws" … add some more here to clarify to what end? E.g. "… in addressing this limitation."

*We will add "…with non-evolving  inputs  to  address this limitation"*

332: "… may vary". Spatially? Temporally? Both?

*Both. We will clarify this.*

Figures

Figure 1 (caption): The meaning of "Annual year start ice fronts" is a little bit unclear here, and an awkward way to start this sentence. Consider revising? "… from SAR intensity images OF Lemos et al …"

*We will reword this*

Figure 4: Clarify in caption what the grey region represents?

This is the period of maximal sliding speed and retreat, and is referenced in Figures 5 and 6. *We will clarify this.*

Figure 5a: Suggest making the symbols in the legend larger (i.e., use filled circles rather than dots). The colored dots on their own there are difficult to see and their colors are difficult to discern from one another (the clouds of dots in the actual figure body are ok as is).

We will increase the circle size.

Figure 8: The actual line colors in the figure here do not seem to agree with the colors as described in the caption. "Vertical-dashed lines indicate the value of u_0". This is confusing since you just said that the colors of the lines (at least for two of them) are for different values of u_0. Do you mean that the dashed-line represents the values assumed over a certain region for the modeling done here?

Thanks for pointing out the colours! I had changed the colour scheme but forgotten to change the caption.

*We will the colour scheme again and update the caption along with it*. The vertical dashed lines just identify the u_0 values for the corresponding curves. The scale in the right hand panel doesn't go up to 10000 m/yr so this value isn't shown. *We will update the caption to better explain this*.